# *DAZL* Knockout Pigs as Recipients for Spermatogonial Stem Cell Transplantation

**DOI:** 10.3390/cells12212582

**Published:** 2023-11-06

**Authors:** Nathalia L. M. Lara, Taylor Goldsmith, Paula Rodriguez-Villamil, Felipe Ongaratto, Staci Solin, Dennis Webster, Uyanga Ganbaatar, Shane Hodgson, Stanislas M. A. S. Corbière, Alla Bondareva, Daniel F. Carlson, Ina Dobrinski

**Affiliations:** 1Faculty of Veterinary Medicine, University of Calgary, Calgary, AB T2N4N1, Canada; nathalia.delimaemart@ucalgary.ca (N.L.M.L.); abondareva@innovatecalgary.com (A.B.); 2Recombinetics, Inc., St. Paul, MN 55121, USA; taylor.goldsmith@mcgill.ca (T.G.); dennis@recombinetics.com (D.W.); hodgson.shane@gmail.com (S.H.); stanislas.corbiere@umontreal.ca (S.M.A.S.C.); dan@recombinetics.com (D.F.C.)

**Keywords:** spermatogonial stem cell transplantation, DAZL, DAZL knockout, genetically modified pigs

## Abstract

Spermatogonial stem cell (SSC) transplantation into the testis of a germ cell (GC)-depleted surrogate allows transmission of donor genotype via donor-derived sperm produced by the recipient. Transplantation of gene-edited SSCs provides an approach to propagate gene-edited large animal models. DAZL is a conserved RNA-binding protein important for GC development, and *DAZL* knockout (KO) causes defects in GC commitment and differentiation. We characterized *DAZL*-KO pigs as SSC transplantation recipients. While there were GCs in 1-week-old (wko) KO, complete GC depletion was observed by 10 wko. Donor GCs were transplanted into 18 *DAZL*-KO recipients at 10–13 wko. At sexual maturity, semen and testes were evaluated for transplantation efficiency and spermatogenesis. Approximately 22% of recipient seminiferous tubules contained GCs, including elongated spermatids and proliferating spermatogonia. The ejaculate of 89% of recipients contained sperm, exclusively from donor origin. However, sperm concentration was lower than the wild-type range. Testicular protein expression and serum hormonal levels were comparable between *DAZL*-KO and wild-type. Intratesticular testosterone and Leydig cell volume were increased, and Leydig cell number decreased in transplanted *DAZL*-KO testis compared to wild-type. In summary, *DAZL*-KO pigs support donor-derived spermatogenesis following SSC transplantation, but low spermatogenic efficiency currently limits their use for the production of offspring.

## 1. Introduction

Spermatogonial stem cell (SSC) transplantation is a technique that holds great potential for addressing male infertility, as well as for endangered species conservation, genetic preservation, and generation of genetically modified animal models [1,2,3,4,5,6,7,8]. SSCs are germline stem cells capable of self-renewal and differentiation into mature sperm, providing the basis of continuous spermatogenesis and male fertility [9,10,11,12,13,14]. When SSCs are transplanted into the testes of infertile males, they can colonize the recipient’s seminiferous tubules and re-establish spermatogenesis, enabling the production of functional sperm and offspring with the donor genetics [15,16,17,18,19,20].

The transplantation of gene-edited germ cells to recipient males is an effective strategy for engineering targeted mutagenesis and offers advantages over traditional methods of generating transgenic models, especially in large animals, such as shortened time and cost efficiency in comparison to cloning or embryo-based technologies, or even by replacing techniques that may be inefficient or unavailable in some species [18,21,22,23,24]. If the genetic change is introduced into the germline before the onset of spermatogenesis, this approach can be advantageous for disease modeling, where gene dosage and epigenetics play a role, or different strains of pigs are required. Therefore, SSC transplantation could serve to accelerate biomedical research, enable efficient propagation of valuable models, and rescue difficult-to-breed genotypes, besides contributing to advancements in human health and the development of novel treatments and therapies [5,25,26]. However, optimization of the transplantation techniques is still ongoing, especially in large animals, where improvement of SSC expansion and survival in vitro, colonization of the recipient testes by transplanted cells, and long-term functionality of the transplanted SSCs are research goals [2,5,7,14,27,28].

To optimize SSC transplantation efficiency, the recipient testis is ideally devoid of germ cells, as the lack of endogenous spermatogenesis creates available niches for transplanted cells to colonize and differentiate [4,29,30]. This can be achieved by using gene-edited animals or by gonadotoxic treatments such as irradiation, heat shock or chemotherapy drugs like busulfan, for example [31,32,33,34]. However, especially in large animals, there are drawbacks of using gonadotoxic treatments, their effects are variable and do not last for the long term, and they carry the risk of systemic toxicity and damage to testicular somatic cells [22,29,31,35,36,37,38,39]. Additionally, the cost and effort involved in treating each individual can be prohibitive [4,18,34]. On the other hand, genetic sterility through gene editing for germline ablation results in a phenotype that is specific to germ cell depletion, and there should be no recovery of endogenous spermatogenesis with time, so that any resulting offspring will only carry the donor genotype [15,16,29,40,41,42]. While well established in rodent models, some of these strategies cannot be easily reproduced in large animals and livestock or have important drawbacks, and genetic sterility of recipients is often considered the most effective alternative for SSC transplantation in large animals. 

Gene-edited pigs with germline ablation via *NANOS2* knockout were previously generated and used as recipients for SSC transplantation with limited success [41,42]. Although donor-derived sperm could be produced, these studies showed low colonization of the porcine testis by the transplanted germ cells, and sperm production was too low to allow transmission of donor genotype through natural breeding. 

DAZL, also known as “Deleted in Azoospermia-like”, is a highly conserved RNA-binding protein [43,44,45] expressed in germ cells within the testes and is required for fertility [46,47,48,49]. DAZL is a key regulator in the spermatogenic process by modulating the expression of genes involved in germ cell determination, survival, and meiosis. DAZL is therefore primarily involved in the maturation of germ cells, promoting the proliferation of spermatogonia and the differentiation of spermatocytes, ultimately leading to the production of functional spermatozoa [48,49,50,51,52,53,54]. Mutations or deficiencies in the human *DAZL* gene have been associated with male infertility and conditions such as azoospermia, where no sperm are present in the ejaculate [55,56,57,58]. As expected, *DAZL*-deficient animals present a germ-cell-depleted phenotype, with defects in germ cell lineage commitment, spermatogonial differentiation, and meiosis, impairing their ability to progress through spermatogenesis and produce sperm [47,48,50,53,54,59,60,61]. 

Therefore, herein we took advantage of previously generated *DAZL*-KO male pigs [21,54] and hypothesized that these could be used as SSC transplantation recipients and enable cost-effective production of biomedical swine models.

## 2. Materials and Methods

### 2.1. Genetically Modified Pigs

*DAZL* knockout (*DAZL*-KO) males were obtained using transcription activator-like effector nuclease (TALEN)-mediated gene editing in pig fibroblasts as described by Tan and colleagues [21]. Briefly, the founder DAZL-KO pigs were developed using TALEN-stimulated homology-dependent repair followed by cloning (somatic cell nuclear transfer). Then, male and female cellular pools consisting of sequence-validated DAZL^+/−^ clones with confirmed mutation were used to generate a breeding herd of DAZL^+/−^ pigs, which was then used when colony expansion and sample collection were necessary. The experimental pigs obtained were a Landrace/Yorkshire crossbreed.

### 2.2. Spermatogonial Stem Cell Transplantation

We performed SSC transplantation into 18 *DAZL*-KO male recipients when they were 10–13 weeks old, with one testis being injected with donor GCs while the contralateral testis was kept intact. Transplantation was performed as previously described, with aseptic surgical conditions and under general anesthesia [18,62]. Germ cells from 9-week-old donors with a variety of genetics and breeds (Table 1) were obtained via a sequential enzymatic digestion procedure [18,63]. The resulting cell suspension was enriched for germ cells via differential plating, as germ cells remain suspended in the culture medium while somatic cells adhere to the culture plates. After three rounds of incubation when supernatants were pooled [63], germ cells were collected, centrifuged, and resuspended in DMEM and injected into the center of the rete testis by gravity flow using ultrasound-guided cannulation and a 20-gauge catheter. After transplantation, recipient boars were raised to sexual maturity and trained for semen collection.

### 2.3. Tissue, Sperm, and Blood Collection

Ejaculates were evaluated according to sperm presence, concentration, and motility by light microscopy. Sperm genomic DNA was differentially extracted to reduce the recipient’s non-sperm cells within the seminal plasma and concentrate the sperm heads. Single-nucleotide polymorphisms (SNP) identified for the recipient tail and donor genomic DNA were PCR-amplified and Sanger-sequenced to confirm the origin (donor/recipient) of the sperm collected (Appendix A). 

Non-transplanted control, heterozygotes, and knockout pigs were castrated at 1 week and 10 weeks of age and at sexual maturity for evaluation of the testicular parenchyma, while transplanted *DAZL*-KO boars were castrated only at sexual maturity, when blood was also collected for hormonal analyses. Blood was allowed to clot for approximately 30 min at room temperature before being centrifuged at 1000× *g* for 10 min at 4 °C. Blood serum was collected, aliquoted, and frozen immediately. For intratesticular testosterone measurements, fragments of the transplanted and non-transplanted testes were homogenized in PBS containing protease inhibitors and detergent according to the service provider guidelines, centrifuged at 10,000× *g* for 10 min at 4 °C, and supernatant was frozen immediately. Samples were submitted to the service provider (Eve Technologies, Calgary, AB, Canada) and analyzed using the “Steroid-Thyroid Hormone 6-Plex Discovery Assay”.

Collected testes were fixed in 2% paraformaldehyde or Bouin’s solution and routinely processed for histomorphometry and immunohistochemistry analyses through dehydration via graded alcohol series and embedding in paraffin. Slides were prepared with sections of 5 μm thickness, stained with hematoxylin and eosin or periodic acid-Schiff, and imaged under light microscopy for histological evaluation and testicular morphometry.

### 2.4. Testis Morphometry

The volume densities of tubular and intertubular components of the testicular parenchyma were determined at 400× magnification using Image J software (version 1.53n) with a 713-point grid over digital images. Ten randomly chosen images (7130 points in total) were counted for each animal, and points were classified as one of the following: seminiferous tubules, comprising tunica propria, seminiferous epithelium, germ cells, and tubular lumen; intertubular compartment, including Leydig cells, connective tissue, and blood vessels. The volume of each component was calculated as the product of its volume density and testis volume, considering 1.0 for the specific density of the testis and excluding the testis capsule from the testis weight [64].

The mean Sertoli and Leydig cell nuclear volumes were determined by measuring the nuclear diameter of 30 nuclei showing an evident nucleolus per animal and the calculation of nuclear volume (µm^3^) according to the formula v = 4/3πr^3^, where r = nuclear diameter/2. The total number of Sertoli cells per testis was determined as follows: total number of Sertoli cells per testis = total volume of Sertoli cells in the testicular parenchyma (mL)/Sertoli cell nuclear volume (µm^3^). The volume of individual Leydig cells was obtained from the measurement of their nuclear and cytoplasmic volume. To calculate the proportion between nucleus and cytoplasm and relative nuclear and cytoplasmic volumes, a 713-point square lattice was projected over digital images at 630× magnification, and 1000 points over Leydig cells were counted for each animal. The total number of Leydig cells per testis was calculated from their individual volume and the volume density occupied by Leydig cells in the testis parenchyma.

Slides from the transplanted testes were prepared with sections from different regions across the sample, aiming to obtain a representation of the whole organ. These sections were digitalized using a Zeiss Axio Scan Z.1 slide scanner at 10× magnification. From the full-section images obtained, all seminiferous tubule cross sections (average of 600 tubules cross sections per slide) were evaluated and categorized into having or not having germ cells in the seminiferous epithelium.

### 2.5. Immunohistochemistry

Testis sections were deparaffinized with xylene and rehydrated. Antigen retrieval was performed by boiling for 10 min in citrate buffer (antigen unmasking solution, Vector, #H-3300-250). Slides were allowed to cool, washed in PBS, and permeabilized in cold methanol for 10 min, followed by 10 min in 0.2% Triton-X in PBS. Slides were then washed in PBS twice and unspecific binding was blocked by incubating with CAS-Block for 30 min (Thermo Fisher Scientific, Waltham, MA, USA, #008120). Sections were then incubated with the primary antibodies (Appendix A) diluted in PBS + 10% CAS-Block overnight at 4 °C. On the next day, slides were washed with 0.05% Tween 20 in PBS three times, incubated with the appropriate secondary antibodies (Appendix A) for 1 h at room temperature, washed with PBS, and mounted with Vectashield with DAPI (Vector, #H-1200-10) for visualization and imaging with a fluorescence microscope.

### 2.6. Statistical Analysis

Statistical analysis was performed using GraphPad Prism v6.0 (GraphPad Software Inc., San Diego, CA, USA). Data were tested for normality (D’Agostino and Pearson) and analyzed accordingly using either one-way ANOVA with Tukey’s multiple comparison tests or Student’s *t*-test. The limit of statistical significance considered was *p* < 0.05 (* *p*  <  0.05, ** *p*  <  0.01, and *** *p*  <  0.001).

## 3. Results

### 3.1. Composition of the Testicular Parenchyma in DAZL-KO Pigs 

We first characterized the testes of heterozygote (Het) and homozygote *DAZL*-KO (KO) pigs at 1 week old (1 wko) and in adulthood. Figure 1 shows that the testicular parenchyma’s general structure in Het and KO pigs was similar to that of wild-type (WT) pigs, with both tubular and intertubular compartments clearly present. Interestingly, gonocytes were present in testes of all genotypes at 1 wko, including in KO testes (arrowheads, Figure 1). However, these germ cells were no longer observed in adulthood in the KO testes, while Het testes showed a mix of Sertoli-cell-only tubules and seminiferous tubules with full spermatogenesis (asterisks, Figure 1). 

To determine an appropriate time point for SSC transplantation, we monitored the presence of germ cells based on the expression of UCH-L1, a marker for gonocytes/early spermatogonia [22,28,34]. While some germ cells were present in 1 wko *DAZL*-KO pig testes, complete germ cell depletion was observed at 10 weeks of age (10 wko) as well as in adult testes (Figure 2). Therefore, we performed transplantation when KO recipients were 10 to 13 weeks old. 

### 3.2. Transplantation Efficiency

Transplantation was performed using germ cells from a variety of donor breeds and genetics (Table 1), as the intended application of the project was to use the SSC transplantation approach to efficiently propagate swine models by transplanting gene-edited germ cells. As there was no effect of donor cell origin and transplantation outcomes, all samples were combined for analysis. An overview of the transplantation parameters, the different donors, the number of germ cells injected, and sperm production efficiency is shown in Table 1. 

The number of germ cells transplanted into each recipient’s testis varied from 26 million to 137 million, but there was no apparent correlation between a higher number of cells injected and increased sperm production. Sperm were observed in the ejaculates of 89% (16 out of 18) of transplanted males. Peak sperm concentration ranged from 5 × 10^4^ to 7.3 × 10^6^ per mL. For comparison, wild-type sperm concentration typically ranges from 1 × 10^9^ to 5 × 10^9^ per mL. We were able to extract sperm genomic DNA from 13 transplanted animals, and SNP analysis (Figure 4c) confirmed that all sperm genotyped were exclusively from donor origin. Therefore, *DAZL*-KO pigs can produce donor-derived sperm after germ cell transplantation. Sperm motility was low, however, as motile sperm were observed in the ejaculates of only 4 of 18 recipients, with peak sperm motility being 0–10% for most recipients, while one recipient showed more consistent motility at 80%.

Figure 3 shows the histology of the *DAZL*-KO transplanted testis at the time of castration. Morphometric quantification revealed that 22.2% (±15.2, *n* = 4) of the seminiferous tubule cross sections contained germ cells at different stages of development, and most germ-cell-containing tubules were observed at the region close to the transplantation site (mediastinum/rete testis). Immunostaining of the recipient testes that received donor germ cells confirmed the presence of elongating spermatids expressing TNP1 (nuclear transition protein 1) (Figure 4a) as well as proliferating spermatogonia expressing UCH-L1 and the proliferating cell nuclear antigen (PCNA) (Figure 4b).

**Figure 3 cells-12-02582-f003:**
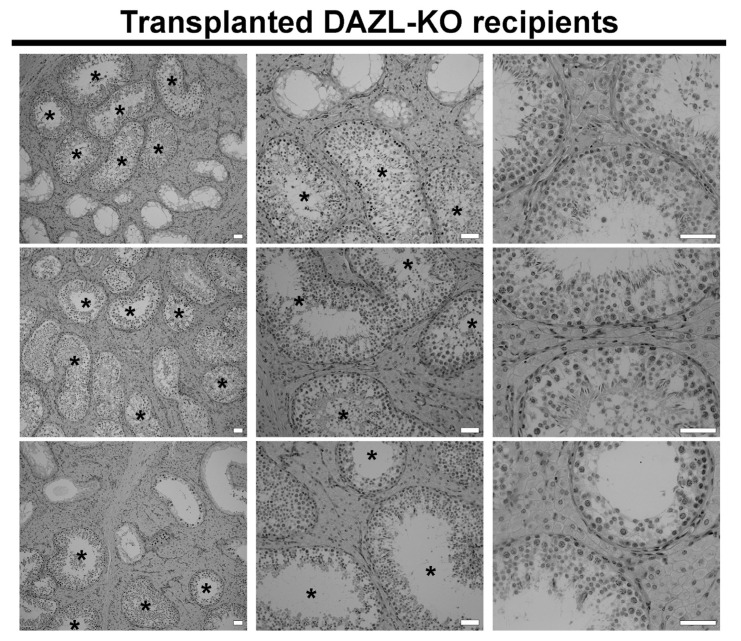
Histology of *DAZL*-KO recipient testes after germ cell transplantation. Histology of the germ cell transplantation recipient testes at the time of castration, at different magnifications. While some seminiferous tubule cross sections are still devoid of germ cells, other tubules contained germ cells at different stages of development (black asterisks). Samples represented are 2906, 2615, and 2626. Scale bars = 50 µm.

**Figure 4 cells-12-02582-f004:**
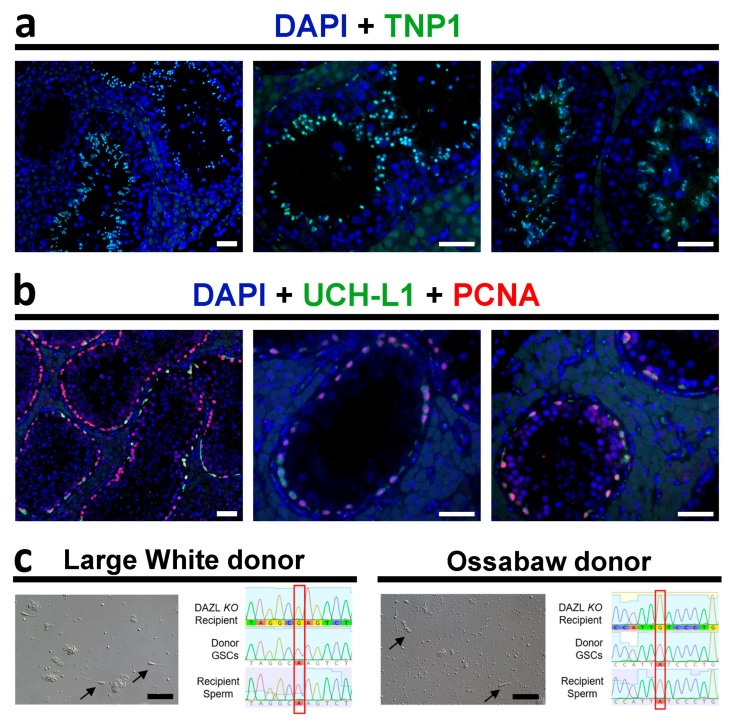
Presence of elongating spermatids and proliferating spermatogonia in transplanted *DAZL*-KO testes and sperm genotyping results. TNP1+ elongating spermatids ((**a**); green) and proliferating spermatogonia ((**b**); UCHL1+ in green and PCNA+ in red) are present in seminiferous tubules cross sections of *DAZL*-KO recipients after germ cell transplantation. (**c**) SNP analysis on sperm genomic DNA extracted from recipient pigs confirmed that all sperm (black arrows) were exclusively from donor origin (red box). Scale bars = 50 µm.

### 3.3. Functionality of the Recipient Testis 

Given the low sperm production efficiency and sperm quality observed, we further evaluated the recipient testes for markers of somatic cell function at 1 wko and in adulthood, as well as the hormonal profile in adult *DAZL*-KO pigs. Table 2 and Figure 5 summarize the histomorphometry results for testis component occupancy and the Sertoli and Leydig cell volumes and numbers.

At 1 wko, the only significant difference observed was the proportion of the testis occupied by connective tissue that was increased in both Het and KO pigs when compared to control (*p* < 0.05, Table 2), as well as the reduction in germ cell occupancy from 6.8% in WT pigs to 0.8% in Het and 0.2% in KO (*p* < 0.05). In the adult pigs, the differences in testis component occupancies were more pronounced. The absence (or reduced number, in the Het) of germ cells led to a smaller proportion of the testis being occupied by seminiferous epithelium and the tubular compartment itself in both Het and KO groups when compared to WT (*p* < 0.05), while the tunica propria occupancy was higher in KO in comparison to WT (*p* < 0.05). A consequence of reduced tubular compartment occupancy is the observed increase in the intertubular compartment proportion in Het and KO pigs (*p* < 0.05), which was accompanied by higher occupancy of both the Leydig cells and connective tissue present in this compartment (*p* < 0.05).

When analyzing Leydig and Sertoli cells specifically (Figure 5), we observed that Leydig cell individual volume was significantly increased (*p* < 0.05; Figure 5a) while the number of these cells per testis was reduced (*p* < 0.05; Figure 5b) in adult KO pigs when compared to WT, while no difference was observed among 1 wko pigs. Sertoli cell nuclear volume was significantly increased (*p* < 0.05) in Het when compared to WT, and in KO when compared to Het and WT at 1 wko (*p* < 0.05), but no difference was observed in mature Sertoli cell nuclear volume in adulthood (Figure 5c). Additionally, no significant changes were observed in Sertoli cell number per testis in any of the ages evaluated (Figure 5d).

We then evaluated the expression of functional markers: vimentin (VIM), an intermediate filament protein present in the cytoskeleton of testicular somatic cells; the Anti-Müllerian Hormone (AMH), produced by immature Sertoli cells; occludin (OCLN), a protein present in tight junctions among adjacent mature Sertoli cells and some interstitial cells; alpha-smooth muscle actin (SMA), present in peritubular myoid cells and endothelial cells; and aromatase (CYP19), a steroidogenic enzyme present in Leydig cells. The expression of these markers in 1 wko (Appendix A) and adult testes (Appendix A) was similar between the WT, Het, and KO pigs and followed the expected localization and age-appropriate patterns.

Lastly, we measured the serum levels of some hormones from blood collected during castration, as well as serum and intratesticular testosterone levels on both transplanted and non-transplanted testes (Figure 6). The serum levels of progesterone (P4), cortisol, T3, and T4 did not change between WT and KO pigs, while estradiol (E2) levels were increased in KO pigs (*p* < 0.05) (Figure 6a). No difference was observed in serum testosterone levels, but intratesticular levels of testosterone were increased in the KO-transplanted testis in comparison to WT (*p* < 0.05) (Figure 6b).

## 4. Discussion

The successful application of the SSC transplantation technique will allow the development of novel strategies in assisted reproduction, preservation of genetic diversity in livestock species, rescue and propagation of difficult-to-breed genotypes, and generation of genetically modified animal models. We previously reported targeted gene editing in porcine germ cells that can serve as donor cells for SSC transplantation and enable the successful generation of gene-edited pig models [65]. In the present study, we focused on investigating the potential use of *DAZL*-KO pigs as hosts for SSC transplantation.

We confirmed the impaired spermatogenesis previously observed in *DAZL*-KO pigs [54], which is consistent with the phenotype described in other species [46,47,50,55,56,57,59,60,61]. Moreover, we showed that neonatal *DAZL*-KO seminiferous cords still harbor gonocytes, but these GCs are no longer present in the testis of knockout pigs at 10 weeks of age and onwards. The exact age and the mechanism by which the postnatal GC development is disrupted in this model were not within the scope of this study, but these aspects certainly deserve further investigation [47,48,59]. The fact that GCs were completely absent in the KO testis at 10 wko was, however, an essential observation for choosing the age to perform SSC transplantation. At this time point, endogenous GCs are absent, leaving open niches, and testicles are still immature and relatively small to enable better distribution and colonization by the injected cells as previously established with WT recipient pigs [18,31]. Therefore, we performed SSC transplantation when recipient pigs were around 10–13 weeks of age, a time span that falls within the proposed “prime window”, which has been described to be prior to puberty [18,42,66].

The majority (~89%) of *DAZL*-KO recipient pigs were able to produce sperm, and all sperm evaluated were exclusively from donor origin. This confirmed that testes of *DAZL*-KO pigs can be hosts for transplanted SSCs and can regenerate donor-cell-derived spermatogenesis. However, the peak sperm concentration was low when compared to controls, and motile sperm were rarely observed except for one recipient which showed better sperm motility. This probably means that this individual had better colonization of the testis parenchyma by donor cells, therefore showing improved sperm parameters. However, at this point, there is no definite explanation for this observation. Based on histological analyses, the degree of colonization obtained was not very high, which could be one reason why fewer sperm were produced. Similar findings have been reported, including when *NANOS2*-KO pigs were used as recipients for SSC transplantation [41,42]. The degree of colonization of the recipient testis is critical to achieving SSC transplantation success and producing sufficient numbers of motile sperm, specifically if natural breeding is targeted [8,17,18,27,66,67,68]. The necessary level of colonization varies depending on species and is especially difficult to achieve in large animals, such as pigs, due to the size of the recipient testis.

Other factors of the microenvironment within *DAZL*-KO testes may not be optimally supportive for GC colonization, maintenance, and differentiation. To investigate this, we evaluated somatic cell function and hormonal homeostasis in these pigs. Most changes observed in the occupancy of the testicular parenchyma components were related to the lack of germ cells inside the seminiferous tubules, which led to a proportional reduction of tubular compartment occupancy and a higher proportion of intertubular compartment. No major change was observed in Sertoli cell number in either age group, while Leydig cells in the adult *DAZL*-KO testes were larger in size but reduced in numbers. We suspect that the changes observed in these steroidogenic cells may be related to the increase in intratesticular testosterone levels in the KO-transplanted testis in comparison to WT, and the increased serum E2 levels in *DAZL*-KO pigs, considering that high testosterone levels can lead to increased aromatization into E2. These results show that the steroidogenic potential of Leydig cells is maintained in *DAZL*-KO pigs and can be activated in the presence of spermatogenic cells, as testosterone is required for the regulation of spermatogenesis at several steps, such as meiosis and spermiogenesis [69,70,71]. Testosterone is also important for the regulation of the blood–testis barrier, which protects haploid cells within the seminiferous tubules. It can be argued that the adequate performance of this barrier after transplantation and SSC colonization is critical and may be even more important in the context of allogeneic or xenogeneic transplantation [7,69,72]. The serum levels of most hormones investigated did not show any significant changes, and the same is true for the expression of somatic cell functional markers, which did not reveal any differences between WT, Het, and KO pig testes and followed the expected patterns. This indicates that, overall, the functionality of the testicular somatic compartment is maintained in *DAZL*-KO pigs.

In summary, in the present study, we showed that *DAZL*-KO pig testes support donor-derived spermatogenesis and produce sperm following SSC transplantation, but low spermatogenic efficiency limits its use for the production of donor-derived offspring. Therefore, the technique needs to be further improved, possibly focusing on increasing the level of GC colonization of the testis parenchyma, by investigating the potential mechanisms involved or stimulating this process via hormonal modulation for example [73]. Currently, low colonization efficiency is possibly the biggest limitation to the success of routinely applying SSC transplantation in livestock breeding. Nevertheless, our findings open new avenues for SSC transplantation research in livestock that may ultimately enable the efficient production of genetically modified animals, preservation of valuable genetic traits, or other applications where assisted reproductive technologies could be used to counterbalance the low spermatogenic efficiency.

## Figures and Tables

**Figure 1 cells-12-02582-f001:**
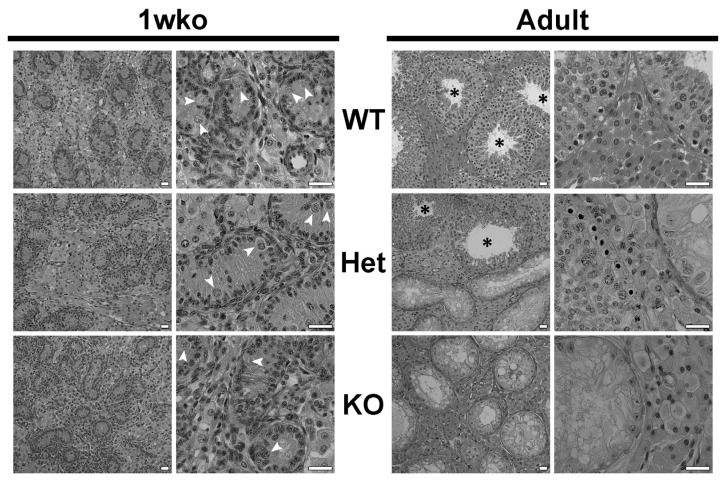
Testicular histology of wild-type (WT), heterozygote (Het), and *DAZL*-KO (KO) pigs at 1 week of age (1 wko) and in adulthood. The histology and organization of the testicular parenchyma in Het and KO testes are similar to the WT. White arrowheads point to gonocytes, present at 1 wko in the testes of all groups, including KO. In adulthood, Het testes show a mix of seminiferous tubules with no germ cells and seminiferous tubules with full spermatogenesis (black asterisks), while no germ cells are observed in the seminiferous tubules of *DAZL*-KO pig testes. Scale bars = 25 µm.

**Figure 2 cells-12-02582-f002:**
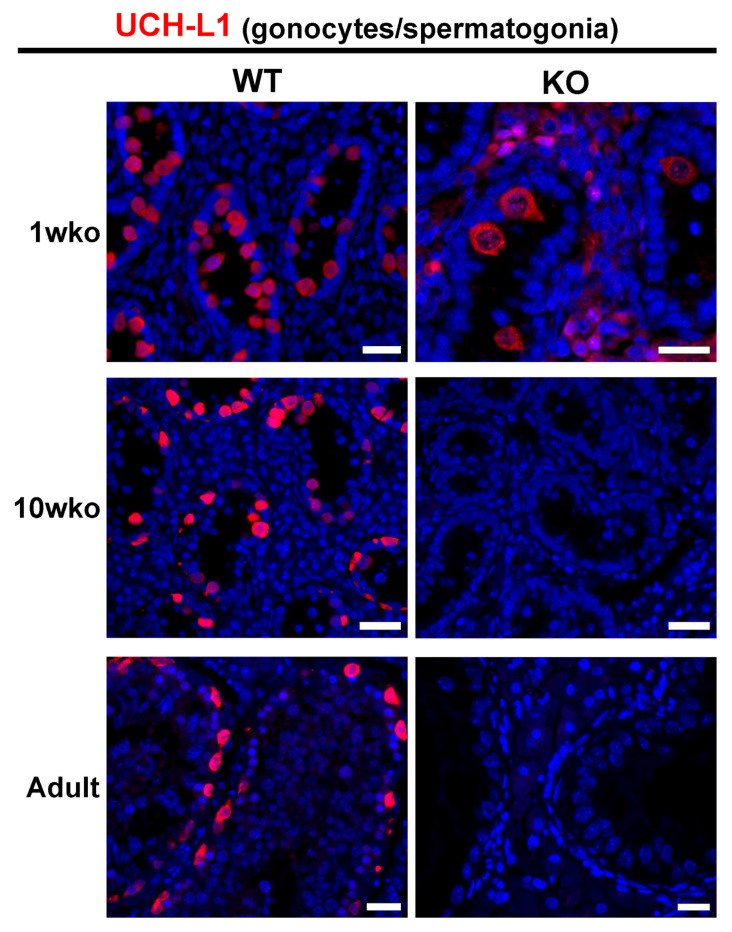
Germ cells (UCHL1+, red) are present in the testis of *DAZL*-KO pigs at 1 wko but no longer present at 10 weeks of age onwards. Some gonocytes were present in 1 wko *DAZL*-KO pig seminiferous cords, but at the time chosen for germ cell transplantation (10 wko), no UCHL1+ germ cells were observed in the KO testis, similar to what is observed in the adult KO testis. Scale bars = 25 µm.

**Figure 5 cells-12-02582-f005:**
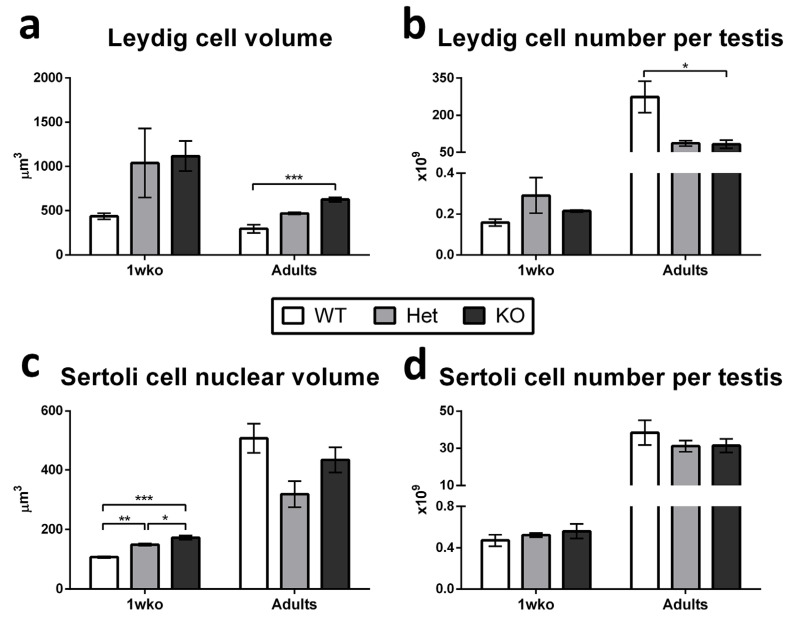
Leydig and Sertoli cell parameters in WT, Het, and *DAZL*-KO pigs testis at 1 wko and in adulthood. (**a**) Leydig cell individual volume is significantly increased (*p* < 0.05) in adult KO pigs in comparison to WT but unchanged among the other groups. (**b**) The number of Leydig cells per testis is decreased (*p* < 0.05) in adult KO pigs in comparison to WT but unchanged among the other groups. (**c**) Sertoli cell nuclear volume was increased in Het when compared to WT (*p* < 0.05) and increased further in KO when compared to Het and KO (*p* < 0.05) at 1 wko, but no difference was observed in mature Sertoli cell nuclear volume in adulthood. (**d**) Sertoli cell number per testis did not change among the groups and ages. Data are shown as mean ± SD and * = *p*  <  0.05, ** = *p*  <  0.01, and *** = *p*  <  0.001.

**Figure 6 cells-12-02582-f006:**
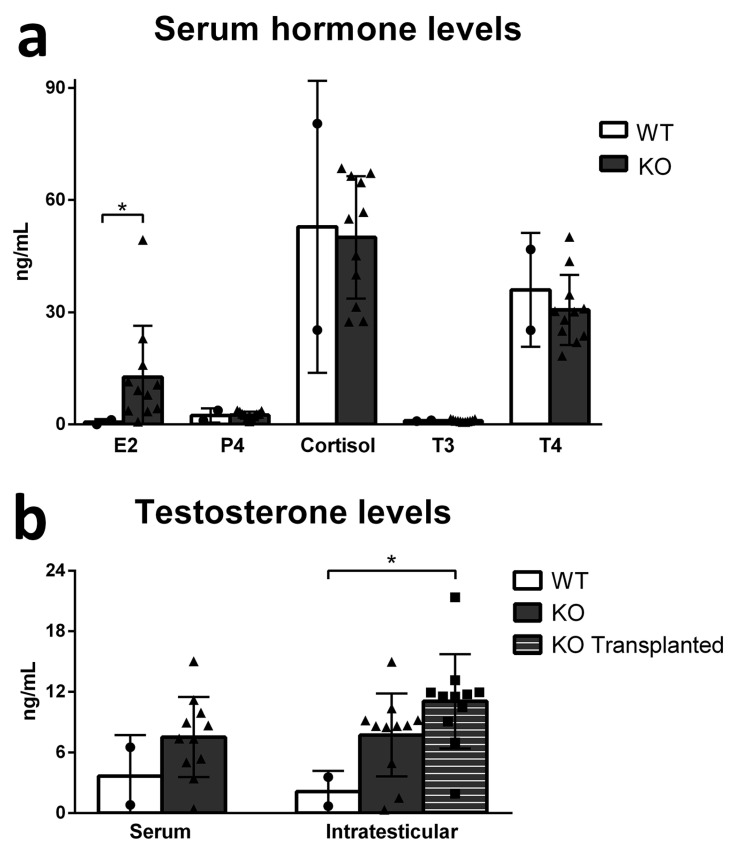
Serum hormone levels and serum and intratesticular testosterone levels in WT (circle) and KO (triangle) pig testis. (**a**) The serum level of estradiol (E2) was higher (*p* < 0.05) in KO pigs, but no change was observed in the levels of progesterone (P4), cortisol, triiodothyronine (T3), and thyroxine (T4). (**b**) The serum testosterone level showed no significant difference between WT and KO, but the intratesticular testosterone level was increased (*p* < 0.05) in the KO testis that received germ cell transplantation (square) in comparison to WT. Data are shown as mean ± SD and * = *p*  <  0.05.

**Table 1 cells-12-02582-t001:** Germ cell transplantation parameters and efficiency.

Recipient ID	Donor Breed	Donor Genetics	GCs Injected (million)	Peak Sperm Concentration/mL	Sperm Genotype
2908	Yorkshire	WT	50	6 × 10^6^	Donor
2906	Yorkshire	Edited	50	4.2 × 10^6^	Donor
3412	Yorkshire	Pooled edited	137.75	Low	Donor
2615	Cross	WT	122	Low	Donor
3327	Cross	WT	70	Low	Donor
3346	Cross	WT	50	2.5 × 10^5^	Donor
3348	Cross	WT	70	5 × 10^4^	Donor
3661	Cross	Pooled WT	68	5.4 × 10^6^	ND
2616	Cross	Edited	93	Low	Donor
3234	Cross	Edited	50	5 × 10^4^	ND
3238	Cross	Edited	50	4 × 10^5^	Donor
3249	Ossabaw	WT	26.46	0	ND
2626	Ossabaw	Pooled WT	34.2	Low	Donor
2988	Ossabaw	Pooled WT	50	7.6 × 10^6^	Donor
3719	Ossabaw	Edited	39.25	7.3 × 10^6^	Donor
3310	Ossabaw	Pooled edited	70	0	ND
5043	Ossabaw	Pooled edited	46.4	1.7 × 10^5^	ND
5047	Ossabaw	Pooled edited	46.4	3 × 10^6^	Donor

Cross = Large White composite. ND = not determined.

**Table 2 cells-12-02582-t002:** Testis component occupancy (volumetric density; %) in wild-type (WT), *DAZL* Heterozygote (Het), and *DAZL* Knockout (KO) 1-week-old (1 wko) and adult pigs (mean ± SEM).

	1 wko			Adults		
	WT	Het	KO	WT	Het	KO
Seminiferous tubules	51.6 ± 4.7	36.2 ± 0.9	39.3 ± 2.5	83.4 ± 3.5 ^a^	66.1 ± 4.6 ^b^	58.4 ± 2.6 ^b^
Seminiferous epithelium	27.6 ± 2.4	25.5 ± 1.1	26.2 ± 1.9	59.9 ± 2.9 ^a^	46.4 ± 1.3 ^b^	33.1 ± 2.3 ^c^
Germ cells	6.8 ± 0.7 ^a^	0.8 ± 0.1 ^b^	0.2 ± 0.1 ^b^	-	-	-
Tubular lumen	-	-	-	17.5 ± 1.2	11.3 ± 4.2	16.1 ± 1.5
Tunica propria	17.2 ± 1.9	9.9 ± 0.1	12.9 ± 0.7	6.0 ± 0.5 ^a^	8.4 ± 0.8 ^ab^	9.2 ± 0.9 ^b^
Intertubular compartment	48.3 ± 4.7	63.8 ± 0.9	60.7 ± 2.5	16.6 ± 3.5 ^a^	33.9 ± 4.6 ^b^	41.6 ± 2.6 ^b^
Leydig cells	19.6 ± 2.9	30.7 ± 0.9	24.3 ± 1.0	9.2 ± 1.6 ^a^	16.9 ± 2.8 ^ab^	22.8 ± 1.4 ^b^
Connective tissue	14.4 ± 0.4 ^a^	21.7 ± 0.3 ^b^	24.2 ± 1.5 ^b^	4.0 ± 0.9 ^a^	10.2 ± 1.4 ^b^	11.6 ± 1.4 ^b^
Blood vessels	14.3 ± 1.7	11.4 ± 0.3	12.2 ± 2.9	3.4 ± 1.2	6.8 ± 0.4	7.2 ± 0.8

Different letters for the same parameter within the same age indicate significant differences (*p* < 0.05).

## Data Availability

The datasets generated and/or analyzed during the current study are available from the corresponding author upon reasonable request.

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
