# Peer review of "DAZL Knockout Pigs as Recipients for Spermatogonial Stem Cell Transplantation"

_cells, 2023, doi:10.3390/cells12212582_

Round 1

Reviewer 1 Report

Comments and Suggestions for Authors

6th October, 2023

Review of Manuscript ID cells-2651753, by Lara N.L.M. et al., entitled: “DAZL knockout pigs as recipients for spermatogonial stem cell transplantation” that is intended for publication in Cells

(The Microsoft Word file as Reviewer Attachment for Manuscript ID cells-2651753 Cells 6th October 2023 has also been added)

The article addresses the transplantation of donor spermatogonial stem cells (SSCs) into the  recipient's testes (young boars with a DAZL gene knockout leading to the elimination of their own germline cells - total or depleted), in the context of assessing their ability to colonize  of the testis parenchyma and resume spermatogenesis. In turn, these processes  resulting in the SSC recipient producing donor SSC sperm. The obtained results provided a number of data that can be useful in designing further research. The idea of transplanting donor SSC into the recipient's depleted testis is attractive, but it seems difficult to translate into an application.

The paper is interesting and well written in English. The text is enriched by carefully designed Figures that well illustrate the contents discussed by the Authors. It is worth highlighting methodological workshop based on the use of a wide variety of methods from the fields of transgenesis, embryology and immunohistochemistry. The Authors have selected the relevant methods for statistical analysis of the results. This enabled to critically evaluate the results achieved by the Authors as compared to the studies of other investigators.

But, in my opinion, the following points should be considered prior to the acceptance of manuscript for publication as has been detailed below:

 1) The Materials and Methods section (subsections 2.1 and 2.2) should be more elaborated and should include some details  such as: concise information on the generation of transgenic boar recipients as well as more details on the preparation of SSCs (whether the cells were cultured, passaged and for how long).

2) The first paragraph of the Discussion is redundant because it duplicates the content of the Introduction.

3) In the Discussion, it is worth trying to explain the curiously high sperm motility (80%) confirmed in one SSC recipient compared to other recipients, whose sperm motility levels oscillate between 0% and 10%.

In conclusion, I recommend this paper for publication in Cells,  after the minor revision of the manuscript will have been completed by the Authors.

Author Response

But, in my opinion, the following points should be considered prior to the acceptance of manuscript for publication as has been detailed below:

1) The Materials and Methods section (subsections 2.1 and 2.2) should be more elaborated and should include some details such as: concise information on the generation of transgenic boar recipients as well as more details on the preparation of SSCs (whether the cells were cultured, passaged and for how long).

Response: We thank the reviewer for pointing it out and we have added more information to these subsections.

2) The first paragraph of the Discussion is redundant because it duplicates the content of the Introduction.

Response: We reviewed the first paragraph of the Discussion, removed the redundant information, and moved the most important parts to the Introduction.

3) In the Discussion, it is worth trying to explain the curiously high sperm motility (80%) confirmed in one SSC recipient compared to other recipients, whose sperm motility levels oscillate between 0% and 10%.

Response: The fact that 1 of the 18 recipients showed better sperm motility truly is curious but as it is only one sample, we can’t really explain what happened other than guessing that the SSC transplantation/donor cell colonization in this individual was more successful, for some unknown reason, in comparison to the usual low efficiency of the transplantation in the other samples. We have added a sentence in the Discussion (Lines 359-363) referring to this.

Reviewer 2 Report

Comments and Suggestions for Authors

In this paper the authors describe the KO DAZL pig model as a possibility for SSC cell implantation. This paper is very interesting and well documented.

I have some questions:

1- Can you explain why different concentrations of GC were injected?

2- What is the normal concentration of spermatozoa in pig semen?

3- You specify that spermatozoids in the ejaculate are few mobile. Have you been able to verify that they are still fertile?

4-Could you standardize the font sizes for the figures?

Author Response

1- Can you explain why different concentrations of GC were injected?

Response: The isolation of germ cells from donor testis is a critical point of our study and a step that is still under optimization in the field of SSC transplantation in large animals, since germ cells are very sensitive to manipulation/culture. Our optimized protocol usually yields approximately 4.5 x 106 cells per gram of donor testis. However, absolute germ cell yield is limited by availability, size and quality of donor testis tissue. In the present study we decided to inject all germ cells obtained after each digestion round, since the recipient testis size was not a limiting factor.

2- What is the normal concentration of spermatozoa in pig semen?

Response: Sperm concentration from wild type boars typically range from 1x109 to 5x109 sperm/mL. We have added this information to the Results section (Lines 228-229).

3- You specify that spermatozoids in the ejaculate are few mobile. Have you been able to verify that they are still fertile?

Response: We did try to achieve donor-derived fertilization by performing IVF using semen from recipient 3719, which showed the best sperm quality. Cleavage and blastocysts rates were 20-35% and 0-9%, respectively, while typical rates for wild-type swine IVF in our lab are ~90% and 25%, respectively, showing lower fertility than conventionally produced semen. Five embryo transfers were performed but pregnancy could not be achieved, and while blastocysts were collected to determine parentage, there were technical issues with the assays and the results were not conclusive. Therefore, we could not get evidence for fertilization and decided not to include this data in the current paper. While we were unable to generate pregnancies in this study, we have advanced the field over previously published germ cell transplantation studies in pigs and gained insight that can lead to future success.

4-Could you standardize the font sizes for the figures?

Response: We have checked and corrected any previous differences in the figures’ font sizes.

Reviewer 3 Report

Comments and Suggestions for Authors

In this study Authors demonstrate the colonisation and differentiation of Spermatogonial stem cell (SSC) in mature sperm when transplanted into the testis of DAZL-KO pigs. DAZL is a conserved RNA binding protein important for germ cells (GC) development. DAZL knockout (KO) animals have defects in GC commitment and differenzation. Donor GCs were transplanted into 18 DAZL-KO recipients at 10-13 wko. At sexual maturity, semen and testes were evaluated for transplantation efficiency and spermatogenesis. Ormonal assessment and Leydig cells volume have been also evaluated. The results show that approximately 22% of recipient seminiferous tubules contained GCs and the ejaculate of 89% of pigs contained sperm, exclusively from donor origin. However, sperm concentration was lower than wild-type range and motile sperm are rarely observed. On the other hand, testicular protein expression and serum hormonal levels were comparable between DAZL- KO and wild-type. Intratesticular testosterone and Leydig cell volume was increased and Leydig cell number decreased in transplanted DAZL-KO testis compared to wild-type. The authors conclude DAZL-KO pigs support donor-derived spermatogenesis following SSC transplantation, but low spermatogenic efficiency currently limits their use for the production of offspring.

Critical points of the study:

  • Small study population (important to increase the study group).
  • Extration of genomic DNA in 12/16 transplanted animals. It cannot be confirmed that all spermatozoa are exclusively of donor origin (it would be important to analyze the genome of all individuals).
  • The presence of factors of the microenvironment within DAZL-KO testes may not be optimally supportive for GCs colonization, maintenance and differentiation. 
  • Low level of GCs colonisation (only 22%). 
Comments on the Quality of English Language

Moderate editing of English language is required.

Author Response

Critical points of the study:

  • Small study population (important to increase the study group).

Response: We have performed transplantation into 18 recipients pigs. Although it would definitely be interesting to increase the sample size due to the variability of sperm quality results, we consider that our study population was satisfactory and good enough to evaluate our data, especially considering the cost, time-span and effort involved in large animals research.

  • Extraction of genomic DNA in 12/16 transplanted animals. It cannot be confirmed that all spermatozoa are exclusively of donor origin (it would be important to analyse the genome of all individuals).

Response: We agree that it would be important to have data from 100% of the transplanted animals, but due to technical reasons some of these results could not be obtained. Still, in all DAZL-KO samples evaluated over time we have not observed the presence of germ cells in non-transplanted testis (after 10 wko). Therefore, we do not believe these recipients are able to produce sperm of their own and any sperm obtained should come from donor cells, as confirmed in 80% of our samples. We have amended the sentence on Line 230 to better represent our findings.

  • The presence of factors of the microenvironment within DAZL-KO testes may not be optimally supportive for GCs colonization, maintenance and differentiation.

Response: We agree with this observation, and it was what led us to evaluate the expression of some important markers and other parameters of testicular function. Although we did not observe any difference in the parameters we evaluated, it is well known that the testicular microenvironment is extremely complex and well-regulated, including the SSC niche. It would be very interesting to perform a more in-depth analysis of more factors that could be involved in GC support within DAZL-KO testes, however it was out of scope of our study.

  • Low level of GCs colonisation (only 22%).

Response: The degree of colonization obtained by us (and previous studies) is indeed low, and this is critical to achieve transplantation success and good sperm production, as we have discussed. However, a good level of colonization is especially difficult to achieve in large animals as the recipient testis is usually very big and the number of donor germ cells being a limiting factor. Further studies focused on improving GC colonization of the testis parenchyma are definitely needed and could improve the efficiency of SSC transplantation in livestock.

Round 2

Reviewer 3 Report

Comments and Suggestions for Authors

No additional comments